# Fairness without Demographics on Electronic Health Records

**Yingtao Luo[1], Zhixun Li[2], Qiang Liu[3,4], Jun Zhu[5]**

[1]Carnegie Mellon University, [2]The Chinese University of Hong Kong
[3]CRIPAC, MAIS, Institute of Automation, Chinese Academy of Sciences
[4]School of Artificial Intelligence, University of Chinese Academy of Sciences
[5]Dept. of Comp. Sci. and Tech., Institute for AI, BNRist Center, Tsinghua-Bosch Joint ML Center, Tsinghua University
yingtaol@andrew.cmu.edu, zxli@se.cuhk.edu.hk, qiang.liu@nlpr.ia.ac.cn, dcszj@tsinghua.edu.cn

## Abstract

Machine learning systems are notoriously prone to biased predictions about certain demographic groups, leading to algorithmic fairness issues. Due to concerns about patient privacy and social inequity, some demographic information may not be available for training a clinical algorithm. Moreover, the complex interaction of different demographics can lead to a lot of unknown minority subpopulations. These challenges greatly limit the applicability of existing group fairness algorithms. To improve the fairness-without-demographics algorithm in the clinical regime, we argue that the gradients of clinical models can provide insights for alleviating inequities. In this paper, we adopt an adversarial weighting architecture and leverage the correlation between model gradients and demographic groups to improve identification and increase exposure of underrepresented groups. We learn the weights of different samples by constructing a graph where samples with similar gradients are connected. Unlike the surrogate grouping methods that cluster groups by proxy sensitive attributes like features and labels, which can be inaccurate, our method provides a soft grouping mechanism that is more robust. The results show that our method can significantly improve fairness without sacrificing too much of the overall accuracy.

## Introduction

Fairness in machine learning has become an urgent concern, as machine learning systems can be biased against certain demographic groups, which contributes to socioeconomic disparities in many areas such as healthcare (Gianfrancesco et al. 2018), finance (Hajian, Bonchi, and Castillo 2016), etc. For example, when learning the risk of patients with different races, due to certain biases, the model prediction can be inaccurate for certain protected groups, such as minorities. To address this issue, most existing methods (Hashimoto et al. 2018; Sagawa et al. 2019; Lahoti et al. 2020; Creager, Jacobsen, and Zemel 2021; Rahman and Purushotham 2022; Chai, Jang, and Wang 2022) require sensitive attributes, such as race, gender, etc., to identify which group is discriminated against by machine learning models. However, due to privacy concerns, these sensitive attributes are not always accessible. For example, regulations like the HIPAA privacy rule have established safeguards to protect

the privacy of health information. In addition, the interaction between various demographic factors can be complex, and the potential protected groups increase exponentially as the number of sensitive attributes increases. This, in turn, escalates the difficulty of identifying the most disadvantaged group. Consequently, it is crucial to advocate for machine learning fairness that does not rely on demographic information.

To ensure fairness without demographics, many existing methods with proxy sensitive attributes (Yan, Kao, and Ferrara 2020; Grari, Lamprier, and Detyniecki 2021; Du et al. 2021; Zhao et al. 2022) assume the correlation between sensitive attributes (groups) and nonsensitive attributes (features), perform clustering to obtain surrogate groups, and enforce group fairness. The problem with these methods, however, is the difficulty in guaranteeing a large overlap with the real protected groups, especially when many protected groups are unknown and the distributional discrepancies between sensitive attributes are large (Chai, Jang, and Wang 2022). Other methods such as ARL (Lahoti et al. 2020) generate weights for different samples, but they can be susceptible to noise (e.g., mislabelling) when outliers are given superior weights due to their rarity in the data and thus may lead to severe degradation of fairness metrics.

In this paper, we develop an innovative adversarial learning framework that comprises a main-task learner and an adversary component tasked with generating sample weights to maximize the learner network's loss. These weights are subsequently utilized in the minimization of the learner's loss. Our approach harnesses the gradients of the learner to categorize samples based on demographics, forming a "Graph of Gradients" (GoG). In this graph, each sample is linked to its K-nearest counterparts exhibiting similar gradient profiles. This method enables the computation of each sample's weight through the aggregation of weights from neighboring samples in the GoG. This soft-grouping mechanism effectively identifies similar samples, while avoiding the imposition of rigid demarcation lines between different demographic groups and preventing the undue influence of noises. Through comprehensive experimental evaluations, we demonstrate that our methodology not only markedly enhances fairness, but also optimizes the balance between fairness and accuracy.

In summary, our contributions are listed as follows.

- We propose a fairness-without-demographics algorithm for clinical models to mitigate the machine learning unfairness issue, which can scale up to large datasets of diverse complexities and structures.
- We show that the gradients of a machine learning clinical model are more effective in representing demographic groups under mild assumption that model accuracy and input features are strongly correlated, which holds true if the deep learning learner is better than random guess. We also show that the last-layer gradients are sufficient.
- We propose to identify demographic subgroups by a soft-grouping method, i.e., graph of gradients (GoG). The proposed method can address the issues with surrogate groups and noisy outliers.
- Extensive experiments on three public datasets and five baselines show that our method outperforms other representative fairness algorithms significantly in terms of both fairness and accuracy.

## Related Work

### Group fairness for classification

Group fairness is a concept that aims to ensure that the outcomes of an algorithm are equitable across different subpopulations defined by sensitive attributes, such as race, gender, etc. To alleviate the group disparity (Jiang et al. 2022), Equal Opportunity (Hardt, Price, and Srebro 2016) hopes that the true positive rates should be the same for all subpopulations, and Predictive Equality (Chouldechova 2017) requires the equality of false positive rates. Preprocessing methods (Chen, Johansson, and Sontag 2018; Jang, Zheng, and Wang 2021) ensure that the data used for training are unbiased and representative of different subgroups by resampling, feature selection, etc. In-processing methods (Madras et al. 2018; Iosifidis and Ntoutsi 2019; Roh et al. 2021; Chai, Jang, and Wang 2022; Chai and Wang 2022) regularize the training process with fair constraints, sample reweighting, and adversarial training. Post-processing methods (Pleiss et al. 2017; Kim et al. 2022; Jang, Shi, and Wang 2022) focus on adjusting the model prediction after training by threshold adjustment, calibration, etc., which are usually very efficient. However, to guarantee group fairness, the availability of sensitive information is a necessity. Some papers (Celis et al. 2021; Celis, Mehrotra, and Vishnoi 2021; Giguere et al. 2022) also address fairness concerns by using techniques that are robust to noisy or shifting sensitive attributes.

### Fairness without demographics

To resolve challenges to discovering the worst-off groups due to both the regulatory limitations and the complex interaction of many demographic variables (Shui et al. 2022), increasing methods are proposed in recent years to achieve fairness without demographics. Some methods follow the Rawlsian Max-Min fairness (Rawls 2004) to minimize the empirical risk of the group with the least utility. For example, Distributionally Robust Optimization (DRO) (Hashimoto et al. 2018) proposes to use $\chi^2$-divergence to discover and minimize the worst-case distribution repeatedly, which essentially only focuses on the learning of the

worst-off group. Adversarial Reweighted Learning (ARL) (Lahoti et al. 2020) uses an adversary network to generate sample weights that maximize the empirical risk and performs weighted learning for the learner model. Based on the concept of computational identifiability, ARL hypothesizes that it can learn demographic information from data features and labels. Surrogate grouping methods (Zhao et al. 2022) are also proposed to minimize the correlation between data features and model prediction, or directly predict surrogate demographic groups (Yan, Kao, and Ferrara 2020) and then perform group fairness algorithms (Sagawa et al. 2019; Rahman and Purushotham 2022). Some debiasing methods propose to identify the group disparities based on clustering information and upsample the minority groups to balance the distribution (Chai, Jang, and Wang 2022; Kim et al. 2022).

## Theoretical Foundation

### Problem Formulation

Consider data $(x, y, a)$ with $n$ samples, where $x$ represents the non-sensitive features, $y$ represents the labels, and $a$ represents the sensitive attributes. Then, given $x$, we need to predict $y$ without the knowledge of $a$ while satisfying certain fairness criteria with respect to $a$. For multi-class classification, $y \in \{M\}$ where $M$ denotes the number of classes.

### Correlation between gradients and demographics

Since we do not have true demographics $a$ as the label, we do not know $a$ as an estimated function of $x$ and $U(h)$ a priori. Therefore, we propose to use gradients to represent demographic groups. Gradients provide not only the data bias but also the model bias. As long as the model prediction error differs in different groups, $U(h)$ and $a$ must be correlated.

Consider a neural network model $h$ parametrized by $\theta$ as $h(x; \theta) = \hat{y}$, where $\theta = (W, V)$. We have $W = (W_1, ..., W_d)^\top \in \mathbb{R}^{D \times M}$ as the weight of the last layer, where $D$ denotes the dimensionality of the last latent representation. $V$ is the weight of all the previous layers. $h(x; \theta) = \sigma(W \times z(x; V))$, where $\sigma(z)_j = e^{z_j} / \sum_{d=1}^{D} e^{z_d}$. The last-layer gradient w.r.t. the cross entropy loss is calculated as

$$\frac{\partial}{\partial W} L(h(x; \theta), y) = z(x; V) \times (\hat{y} - y), \quad (1)$$

where

$$L(h(x; \theta), y) = -\sum_d y_d \cdot \log(h(x; \theta)) \quad (2)$$

$$= \log \left( \sum_{d=1}^{D} e^{W_d \cdot z(x; V)} \right) - W_y \cdot z(x; V). \quad (3)$$

Note that $\hat{y} - y$ is the bias of the model prediction, which can have a positive/negative value. We define the undirected gradient $g \in \mathbb{R}^{D \times M}$ of the last layer of $h$ by

$$g_{d,j} = z(x)_d |\hat{y}_j - y_j| = z_d U_j, \quad (4)$$

which is the latent representation multiplication of non-sensitive feature and prediction error. Here, $y_j$ denotes the

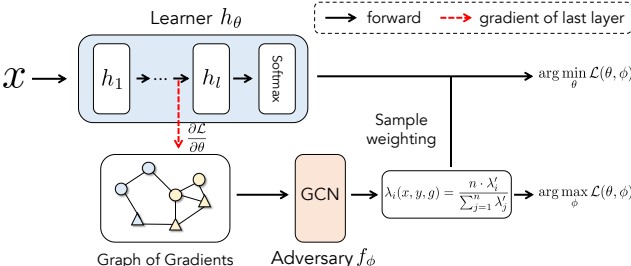

Figure 1: The proposed Graph of Gradients framework.

true value of the $j$-th class on the label. We show that using the undirected gradient to estimate sensitive attributes is more accurate than using only $x$ by Theorem 1.

**Theorem 1.** *The distribution of gradients has a closer distance to sensitive attributes than input features. If we denote sensitive demographics as $Z$, model prediction error as $Y$, and input features as $X$, $I(XY|Z) > I(X|Z)$.*

**Lemma 1.** *If we denote sensitive demographics as $Z$, model prediction error as $Y$, and input features as $X$, $\frac{Corr(XY,Z)}{Corr(X,Z)}$ increases when $Corr(X,Y)$ increases.*

**Proposition 1.** *The last-layer gradient of the deep learning prediction model can have a stronger correlation to sensitive attributes than non-sensitive input features. If we denote input features as $x$, model prediction error as $U$, last-layer representation as $z$, and sensitive attribute classes as $s$, we have $Corr(zU, s) > Corr(x, s)$.*

The proofs of Theorem 1, Lemma 1 and Proposition 1 are presented in Appendix. By Proposition 1, we show that the last-layer gradient of a deep learning model is more effective in identifying the demographic groups than the commonly used non-sensitive input features. Based on this property, we develop an architecture in the next section to improve fairness without demographics.

## Methodology

In the literature (Lahoti et al. 2020), it is proven that the Rawlsian Max-Min fairness objective (Rawls 2004) can be formulated as

$$h^* = \arg \min_\theta \max_\lambda \sum_{i=1}^n \lambda_{s_i} L(h(x_i; \theta), y_i). \qquad (5)$$

where $n$ denotes the number of samples. The optimal hypothesis $h^*$ is replaced by a model, and the demographic group $s$ that minimizes the utility is replaced by reweighting each group with a learning weight $\lambda_s$. When the demographic groups are unknown to us, people learn to predict $s(x, y)$, or simply assign weights $\lambda_i$ to each sample.

In this paper, we propose a novel learning framework to address the deficiencies of existing approaches. The overall framework is shown in Figure 1. As hard boundaries for group partition can be intractable, we propose a method that mimics the grouping effect to alleviate the noises. In detail,

our method can reformulate Eq. 7 as follows

$$J(\theta, \phi) = \min_\theta \max_\phi \sum_{i=1}^n \lambda_i(x, y, g; \phi) \cdot L(h_\theta(x_i), y_i), \quad (6)$$

where $\lambda$ is an adversary network powered by graph convolutional network and $g$ is the undirected last-layer gradients of all samples. $L$ is the cross-entropy loss as introduced in Eq.3. $x = [x_1, ..., x_n], y = [y_1, ..., y_n]$. For $g = [g_1, ..., g_n]$, the calculation of $g_i \in \mathbb{R}^{DM}$ for each sample $i \in \{n\}$ is

$$g_i = \text{Flatten} \left( z(x_i) \times |h(x_i) - y_i| \right), \qquad (7)$$

where $h(x_i) \in \mathbb{R}^M$ denotes the prediction of the learner model and $y_i \in \mathbb{R}^M$ denotes the true label. $z(x_i) \in \mathbb{R}^D$ denotes the latent representation of the learner model before the last layer. The calculation of $\lambda$ is

$$\lambda'(x, y, g) = f_\phi(H, A) \in \mathbb{R}^n, \qquad (8)$$

$$\lambda_i(x, y, g) = \frac{n \cdot \lambda'_i}{\sum_{j=1}^n \lambda'_j}, \qquad (9)$$

where $H = E_g + E_x$. Here, $E_g = W^{(0)} g \in \mathbb{R}^{n \times r}$ is the embedding of gradients, with $W^{(0)} \in \mathbb{R}^{DM \times r}$. $E_x = W^{(1)} x \in \mathbb{R}^{n \times r}$ is the data embedding, with $W^{(1)} \in \mathbb{R}^{t' \times r}$ where $t'$ denotes the number of dimensions of $x$. We use $\lambda_i \in \lambda$ to represent the weight for sample $i$, which is normalized according to Eq. 11. On the other hand, $A \in \mathbb{R}^{n \times n}$ is the adjacency matrix for constructing the Graph of Gradients (GoG), which is calculated as follows for a certain entry $A_{u,i}$

$$A_{u,i} = \begin{cases} 1 & \text{if } dist(g_u, g_i) \geq dist(g_u)_k \\ 0 & \text{if } dist(g_u, g_i) < dist(g_u)_k \end{cases}, \qquad (10)$$

where $dist(\cdot, \cdot)$ is the Euclidean distance between the gradients of two samples, $dist(\cdot)_k$ denotes the distance between the gradient of the sample and the gradient of its K-th nearest neighbors among the gradients of other samples.

The calculation of $f_\phi$ can be represented as

$$f(H, A) = \sigma(AHW^{(2)}) \in \mathbb{R}^{n \times 1}, \qquad (11)$$

which is a one-layer graph convolutional network (GCN) that takes each sample's data as input and outputs the sample weight. Therefore, $\phi = [W^{(0)}, W^{(1)}, W^{(2)}]$. To generate the weights, each sample, which is a node in the graph, aggregates the information of similar samples with K-nearest gradients. Here, $W^{(2)} \in \mathbb{R}^{r \times 1}$ allows the graph network to learn the importance of neighboring samples for aggregation, and $\sigma$ denotes the activation function.

The learning of GoG is a grouping mechanism without knowing the demographic groups a priori. The overall algorithm is shown in Algorithm 1 in Appendix.

## Experiments

In this section, we conduct extensive experiments to verify the effectiveness of our method.

## Experimental Setup

We randomly divide each dataset by samples into the training, validation, and testing sets in a 0.75:0.1:0.15 ratio. Different from previous works such as (Lahoti et al. 2020) that only consider race and gender, we consider more subpopulation groups to test the fairness of models under a more severe environment. We tune all the hyperparameters with an appropriate range to obtain the optimal evaluation on the validation set for each model. The range of learning rate is {1e-2, 3e-3, 1e-3}, batch size is {16, 32, 64, 128}, hidden dimension is {16, 32, 64}, dropout rate is {0,1, 0.5}. The algorithm will stop if the accuracy of the worst group validation metrics does not increase in twenty epochs, and the test performance will be recorded. All results are averaged under five random seeds. More experimental details are presented in Appendix.

## Datasets

We evaluate our proposed method on the following real-world datasets: MIMIC-III, MIMIC-IV.

- **MIMIC-III Dataset** The Medical Information Mart for Intensive Care database[1] (Johnson et al. 2016) contains the information of patients who stayed in the critical care units of the Beth Israel Deaconess Medical Center between 2001 and 2012. There are 53423 patients and 651048 diagnosis codes recorded in the dataset. The goal is to predict future diagnoses for multi-class classification. Religion, sex, age, and race are protected attributes.

- **MIMIC-IV Dataset** The MIMIC-IV dataset[2] contains patients who stayed in the critical care units of the Beth Israel Deaconess Medical Center between 2008 and 2019. Patients who had less than three admission records are excluded. The average number of visits for the 10023 selected patients is 4.64, the average number of codes in a visit is 14.12, the total number of unique ICD-9 codes in diagnoses/procedures is 6274/1973.

## Performance Comparison

Overall, Table 1 show that our algorithm can significantly improve the fairness of machine learning diagnosis prediction models significantly. The relative increases of Worst-Group Accuracy to the second-best approach by our algorithm are 2.904% on MIMIC-III dataset and 3.769% on MIMIC-IV dataset. In Equalized Odds, the relative improvements by our algorithm are 27.36% on MIMIC-III dataset and 26.93% on MIMIC-IV dataset. In disparate impact, the relative improvements by our algorithm are 25.90% on MIMIC-III dataset and 28.22% on MIMIC-IV dataset. We observe that FairRF generally performs worse than DRO and ARL in Worst-Group Accuracy, but outperforms them in Equalized Odds and Disparate Impact. Compare to our model, DRO and ARL are much less effective in promoting fairness without demographics. It should be noted that group disparity focuses on minimizing the discrepancies between

---

[1]https://physionet.org/content/mimiciii/1.4/
[2]https://physionet.org/content/mimiciv/0.4/

---

Table 1: Performances of fairness algorithms on the MIMIC-III and MIMIC-IV datasets, evaluated by Acc@20 of the worst group, Equalized Odds, and Disparate Impact. Each result is averaged over ten random seeds.

| Approach | MIMIC-III Dataset | | |
| | W. Acc($\uparrow$) | E. Odds($\downarrow$) | D. Impact($\downarrow$) |
| --- | --- | --- | --- |
| RETAIN | 0.2102± 0.014 | 29.28± 0.45% | 23.89± 0.59% |
| +DRO | 0.2188± 0.034 | 27.75± 0.91% | 23.02± 0.75% |
| +ARL | 0.2187± 0.030 | 27.14± 0.90% | 22.90± 0.77% |
| +FairRF | 0.2169± 0.024 | 26.05± 0.70% | 22.76± 0.66% |
| +Ours | **0.2226** ± 0.034 | **18.99** ± 0.72% | **17.35** ± 0.61% |
| Dipole | 0.1943± 0.032 | 30.09± 0.66% | 24.03± 0.45% |
| +DRO | 0.1979± 0.040 | 28.10± 0.87% | 23.02± 0.72% |
| +ARL | 0.1988± 0.049 | 27.60± 0.85% | 23.11± 0.60% |
| +FairRF | 0.1970± 0.037 | 26.63± 0.65% | 22.79± 0.53% |
| +Ours | **0.2095** ± 0.035 | **19.32** ± 0.72% | **16.84** ± 0.55% |
| Stagenet | 0.2086± 0.012 | 29.53± 0.22% | 23.16± 0.27% |
| +DRO | 0.2127± 0.022 | 27.99± 0.49% | 22.63± 0.43% |
| +ARL | 0.2136± 0.026 | 27.89± 0.49% | 22.77± 0.59% |
| +FairRF | 0.2111± 0.015 | 26.36± 0.19% | 22.32± 0.24% |
| +Ours | **0.2170** ± 0.011 | **19.10** ± 0.27% | **16.11** ± 0.37% |

| Approach | MIMIC-IV Dataset | | |
| | W. Acc($\uparrow$) | E. Odds($\downarrow$) | D. Impact($\downarrow$) |
| --- | --- | --- | --- |
| RETAIN | 0.3102±0.018 | 25.64±0.23% | 23.39±0.33% |
| +DRO | 0.3153±0.046 | 23.33±0.58% | 21.27±0.62% |
| +ARL | 0.3147±0.043 | 23.00±0.51% | 21.55±0.45% |
| +FairRF | 0.3126±0.020 | 22.20±0.10% | 20.97±0.28% |
| +Ours | **0.3299**±0.044 | **16.95**±0.67% | **15.32**±0.44% |
| Dipole | 0.3051±0.032 | 29.11±0.36% | 24.75±0.21% |
| +DRO | 0.3077±0.040 | 25.45±0.38% | 22.23±0.20% |
| +ARL | 0.3093±0.049 | 25.19±0.52% | 22.11±0.39% |
| +FairRF | 0.3047±0.037 | 24.30±0.38% | 21.45±0.24% |
| +Ours | **0.3212**±0.035 | **17.61**±0.49% | **15.27**±0.33% |
| Stagenet | 0.3048±0.032 | 28.19±0.17% | 22.95±0.21% |
| +DRO | 0.3089±0.040 | 24.48±0.33% | 22.14±0.39% |
| +ARL | 0.3112±0.049 | 24.22±0.28% | 22.04±0.40% |
| +FairRF | 0.3079±0.037 | 22.40±0.11% | 21.37±0.21% |
| +Ours | **0.3200**±0.025 | **15.77**±0.14% | **15.19**±0.14% |

groups when making predictions, while the worst group accuracy focuses on ensuring that the model is accurate even for the worst-off group.

## Conclusion

In this study, we address fairness concerns in machine learning models for electronic health records, focusing on the challenges posed by the complex interplay of demographic variables and regulatory constraints, which often render demographic information unknown. We present a novel approach to tackle the limitations of existing methods. Specifically, we highlight the importance of gradients to identify subpopulations, and propose to create a graph of gradients by connecting each sample to its K-nearest neighbors. Graph neural networks are adopted to identify demographic groups and generate sample weights. Experimental results reveal that our method significantly enhances the machine learning fairness on electronic health records data.

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

# Appendix

---

**Algorithm 1:** Fairness without demographics with the graph of gradients.

---

**Require:** The labelled data $x = [x_1, x_2, ..., x_t]$ and $y = [y_1, y_2, ..., y_t]$, the demographic sensitive attributes $a = [a_1, a_2, ..., a_t]$; the learner model $h_\theta$, the adversary model $\lambda_\phi$.

1: **while** not convergence **do**
2:     **for** i=0:n **do**
3:         Compute and record $h_\theta(x_i)$, $z(x_i)$ and $g_i$ by Eq.9;
4:         Compute and record $L(h(x_i; \theta), y_i)$ by Eq.3;
5:     **end for**
6:     Compute $H$, $A$ and $\lambda_\phi(x, y, g)$ by Eqs.(10-14);
7:     Compute $J(\theta, \phi) = \sum_{i=0}^n \lambda(x, y, g; \phi)_i \cdot L(h(x_i; \theta), y_i)$;
8:     Fix $\theta$ and optimize $\phi$ by maximizing $J(\theta, \phi)$;
9:     Fix $\phi$ and optimize $\theta$ by minimizing $J(\theta, \phi)$;
10: **end while**
11: **return** $h_\theta$, which will be a fair model.

---

## Experimental Setup Details

### Baseline Models

- **RETAIN:** (Choi et al. 2016) A two-level neural model based on reverse time attention for healthcare.

- **Dipole:** (Ma et al. 2017) An attention-based bidirectional recurrent neural network for healthcare.

- **StageNet:** (Gao et al. 2020) A deep learning model with stage-aware LSTM and convolutional modules for health risk prediction.

For each baseline model, we incorporate it with several fairness algorithms, denoted as $+(FairModel)$.

We compare our method that uses the graph of gradients with the following fairness-without-demographics models that use features for weight generation or clustering.

- **DRO**: (Hashimoto et al. 2018) A fair algorithm that uses $\chi^2$-divergence to discover and minimize the worst-case distribution repeatedly.

- **ARL**: (Lahoti et al. 2020) A fair algorithm that leverages computational identifiability to learn the demographics from features/labels for the Max-Min fairness objective.

- **FairRF**: (Zhao et al. 2022) A fair algorithm that minimizes the correlation between data features and model predictions with importance weighting.

### Evaluation Metrics

**Worst-Group Accuracy** We adopt the top-$k$ accuracy of the worst subpopulation group (W. Acc.), which indicates the model performance under the worst case. Accuracy@$k$ is the same as used in previous works (Ma et al. 2017; Choi et al. 2017) is defined as the correct medical ICD-9 codes ranked in the top $k$ divided by $\min(k, |y_t|)$, where $|y_t|$ is the number of ICD-9 codes in the $(t+1)$-th visit. Here we use $k = 20$.

Table A1: Ablation study of our model, evaluated by Worst-Group Acc@20, Equalized Odds, and Disparate Impact.

| Dataset | Baseline | Approach | W. Acc($\uparrow$) | E. Odds($\downarrow$) | D. Impact($\downarrow$) |
|---|---|---|---|---|---|
| MIMIC-III | RETAIN | Ours | **0.2226**±0.0034 | **18.99**±0.72% | 17.35±0.61% |
| MIMIC-III | RETAIN | -Graph | 0.2193±0.0025 | 24.67±0.56% | 20.28±0.27% |
| MIMIC-III | RETAIN | -Grad | 0.2211±0.0021 | 20.68±0.42% | 19.14±0.26% |
| MIMIC-III | Dipole | Ours | **0.2095**±0.0035 | 19.32±0.72% | 16.84±0.55% |
| MIMIC-III | Dipole | -Graph | 0.1993±0.0031 | 19.67±0.41% | 14.28±0.35% |
| MIMIC-III | Dipole | -Grad | 0.2058±0.0024 | 19.68±0.29% | 13.62±0.27% |

**Equalized Odds** We use the equalized odds (E. Odds), which requires the probability of instances with any two protected attributes $i$, $j$ being assigned to an outcome $k$ are equal, given the label. As there are as many as $M$ different labels, we simply cluster labels into eighteen diagnosis categories based on the ICD-9 code categories[3]. Since there are different demographic groups $S$, we calculate

$$\triangle_{EO} = \sum_{i,j} |E(\hat{y}|S=i, y=k) = E(\hat{y}|S=j, y=k)|.$$
(12)

**Disparate Impact** We use the disparate impact (D. Impact), which requires the prediction to be fair across different groups. This metric may not make sense for medicine and should use with caution, since the prevalence of a disease can indeed be affected by the demographics

$$\triangle_{DP} = \sum_{i,j} |E(\hat{y}|S=i) = E(\hat{y}|S=j)|. \quad (13)$$

## Experimental Ablation Study

We conduct an ablation study in Table A1 to fully understand the effectiveness of different parts in our method. In total, our method consists at least of 1) the construction and learning of the EHR patient graph and 2) the use of gradients to replace features in representing unknown demographics. We report the fairness metrics when we deduct a certain part from the proposed model to evaluate whether it is important. It can be concluded that both parts are useful, as the model without graph or without gradients can both outperform other fairness algorithms. The graph learning plays a more important role in improving fairness.

## Effectiveness of Gradients

In this section, we discuss and theoretically analyze the effectiveness of gradients to represent sensitive demographics. We first generally demonstrate, through the lens of information theory and as articulated in Theorem 1, that the distribution of gradients is more closely aligned with sensitive demographic attributes compared to input features, if input features are not perfect solutions for identifying demographics. Furthermore, we explore under the condition of linear relationships, as outlined in Lemma 1, that model gradients are more effective than input features in a larger correlation between input features and model prediction error.

---

[3]https://en.wikipedia.org/wiki/List_of_ICD-9_codes

**Theorem 1.** *The distribution of gradients has a closer distance to sensitive attributes than input features. If we denote sensitive demographics as $Z$, model prediction error as $Y$, and input features as $X$, $I(XY|Z) > I(X|Z)$.*

*Proof.* To calculate the mutual information $I(XY|Z)$ between $XY$ and $Z$, where $XY$ is the undirected gradient as shown in Eq. 4, we have

$$I(XY|Z) = H(XY) - H(XY|Z). \quad (14)$$

Similarly, we can calculate the mutual information $I(X|Z)$ between $X$ and $Z$ as

$$I(X|Z) = H(X) - H(X|Z). \quad (15)$$

Subtracting the above two equations, as long as $X$ and $Z$ are not perfectly dependent with each other, we have

$$
\begin{aligned}
&I(XY|Z) - I(X|Z) \\
&= (H(XY) - H(X)) - (H(XY|Z) - H(X|Z)) \\
&= H(Y|X) - H(Y|XZ) \\
&> 0.
\end{aligned} \quad (16)
$$

$\square$

**Lemma 1.** *If we denote sensitive demographics as $Z$, model prediction error as $Y$, and input features as $X$, $\frac{Corr(XY,Z)}{Corr(X,Z)}$ increases when $Corr(X,Y)$ increases.*

*Proof.* Here, we assume the linearity of the data pattern for simplicity of explanation. We hypothesize that there is a correlation between $X$ and $Z$, and also a correlation between $Y$ and $Z$. For simplicity, we assume $Z = aX + \epsilon_a$, where $\epsilon_a$ is a noise term that represents the part of $Z$ that is uncorrelated to $X$. For simplicity, we can assume $\epsilon_a \sim N(\mu_a, \sigma_a^2)$. Similarly, we have $Z = bY + \epsilon_b$ where $\epsilon_b \sim N(\mu_b, \sigma_b^2)$. We regard $a$ and $b$ as two constants, while the two noise terms $\epsilon_a$ and $\epsilon_b$ are unknown and statistically independent. We assume that $X$ follows the standard normal distribution after preprocessing, thus $\mu_X = 0, \sigma_X^2 = 1$.

Note that we can rearrange the given equalities as follows: $Y = \frac{Z - \epsilon_b}{b}$ and $X = \frac{Z - \epsilon_a}{a} \to Y = \frac{Z - \epsilon_b}{b} = \frac{a}{b}X + \frac{\epsilon_a - \epsilon_b}{b}$ and $XY = \frac{a}{b}X^2 + \frac{\epsilon_a - \epsilon_b}{b}X$.

The covariance between $X$ and $Z$ is

$$
\begin{aligned}
Cov(X, Z) &= Cov(X, aX + \epsilon_a) & (17) \\
&= a \cdot Cov(X, X) + Cov(X, \epsilon_a) & (18) \\
&= a \cdot Var(X) + 0 & (19) \\
&= a, & (20)
\end{aligned}
$$

where $Cov(X, \epsilon_a) = 0$ when the noise term is independent of the feature.

The correlation coefficient between $X$ and $Z$ is

$$
\begin{aligned}
Corr(X, Z) &= \frac{Cov(X, Z)}{\sqrt{Var(X)}\sqrt{Var(Z)}} & (21) \\
&= \frac{a}{\sqrt{a^2 + \sigma_a^2}}. & (22)
\end{aligned}
$$

Similarly, we can calculate

$$Cov(X, Y) = E[X(\frac{a}{b}X + \frac{\epsilon_a - \epsilon_b}{b})] - 0 = \frac{a}{b}, \quad (23)$$

$$Corr(X, Y) = \frac{a}{\sqrt{a^2 + \sigma_a^2 + \sigma_b^2}}, \quad (24)$$

$$Var(Z) = Var(aX + \epsilon_a) = a^2 + \sigma_a^2, \quad (25)$$

$$Cov(Y, Z) = Cov((1/b)(Z - \epsilon_b), Z) = \frac{a^2 + \sigma_a^2}{b}, \quad (26)$$

$$Var(Y) = Var(\frac{a}{b}X) + Var(\frac{\epsilon_a - \epsilon_b}{b}) \quad (27)$$

$$= \frac{a^2 + \sigma_a^2 + \sigma_b^2}{b^2}, \quad (28)$$

$$Corr(Y, Z) = \frac{Cov(Y, Z)}{\sqrt{Var(Y)}\sqrt{Var(Z)}} \quad (29)$$

$$= \frac{\sqrt{a^2 + \sigma_a^2}}{\sqrt{a^2 + \sigma_a^2 + \sigma_b^2}}. \quad (30)$$

From observation, we find that when $\sigma_a^2 + \sigma_b^2 = 0$, $Corr(X, Y) = Corr(X, Z) = Corr(Y, Z) = 1$, regardless of the value of $a, b, \mu_a, \mu_b$. Since we regard $a$ as a constant that is not subject to change, we can conclude that $\sigma_a^2$ and $\sigma_b^2$ can directly determine the correlation.

We have the correlation between $XY$ and $Z$ as

$$Cov(XY, Z) = Cov\left(\frac{a}{b}X^2 + \frac{\epsilon_a - \epsilon_b}{b}X, Z\right) \quad (31)$$

$$
\begin{aligned}
&= \frac{a}{b}Cov(X^2, aX + \epsilon_a) \\
&\quad + \frac{1}{b}Cov((\epsilon_a - \epsilon_b)X, aX + \epsilon_a) \quad (32)
\end{aligned}
$$

$$
\begin{aligned}
&= \frac{a^2}{b}Cov(X^2, X) + \frac{1}{b}Cov((\epsilon_a - \epsilon_b)X, \epsilon_a) \\
&\quad + \frac{1}{b}Cov((\epsilon_a - \epsilon_b)X, aX) \quad (33)
\end{aligned}
$$

$$
\begin{aligned}
&= \frac{a^2}{b}(E[X^3] - E[X]E[X^2]) \\
&\quad + \frac{aE(\epsilon_a - \epsilon_b)E(X^2)}{b} \quad (34)
\end{aligned}
$$

$$= \frac{a(\mu_a - \mu_b)}{b}, \quad (35)$$

where, by the moments of standard normal distribution, $E(X^4) = 3$, $E(X^3) = 0$, and $E(X^2) = 1$.

Then, we compute

$$Var(XY) = Var\left(\frac{a}{b}X^2 + \frac{\epsilon_a - \epsilon_b}{b}X\right) \tag{36}$$

$$= \frac{a^2}{b^2}Var(X^2) + \frac{1}{b^2}\text{Var}(\epsilon_a - \epsilon_b)\text{Var}(X) \tag{37}$$

$$= \frac{a^2}{b^2}Var(X^2) + \frac{1}{b^2}(\text{Var}(\epsilon_a) + \text{Var}(\epsilon_b) - 2\text{Cov}(\epsilon_a, \epsilon_b)) \tag{38}$$

$$= \frac{2a^2 + \sigma_a^2 + \sigma_b^2}{b^2} \tag{39}$$

where $Var(X^2) = E(X^4) - E(X^2)^2 = 3 - 1 = 2$.

Therefore, the correlation coefficient between $XY$ and $Z$ is

$$Corr(XY, Z) = \frac{Cov(XY, Z)}{\sqrt{Var(XY)}\sqrt{Var(Z)}} \tag{40}$$

$$= \frac{a(\mu_a - \mu_b)}{b\sqrt{a^2 + \sigma_a^2}\sqrt{\frac{2a^2 + \sigma_a^2 + \sigma_b^2}{b^2}}}, \tag{41}$$

Comparing $Corr(X, Z)$ and $Corr(XY, Z)$, we have

$$Ratio = \frac{Corr(XY, Z)}{Corr(X, Z)} = \frac{\mu_a - \mu_b}{\sqrt{2a^2 + \sigma_a^2 + \sigma_b^2}}. \tag{42}$$

We can tell that both $Corr(X, Y)$ and $Ratio$ are directly dependent on and decrease in $\sigma_a^2 + \sigma_b^2$. Therefore, $\frac{Corr(XY,Z)}{Corr(X,Z)}$ increases in $Corr(X, Y)$. $\square$

Here, linear correlation is used as an example to illustrate our point, since nonlinear correlation is much harder to measure and analyze. In particular, without further assumptions or knowledge about the data, there could be many possible nonlinear relationships (e.g., logarithmic, polynomial, exponential) and nonlinear correlation measurements (e.g., Hilbert-Schmidt Independence Criterion, Mutual Information, Maximal Information Coefficient).

## Effectiveness of Last-Layer Gradients

Then, we extend it to Proposition 1 to show that when the model is a neural network, as a special case of Theorem 1, using the last-layer gradient is sufficient.

**Proposition 1.** *The last-layer gradient of the deep learning prediction model can have a stronger correlation to sensitive attributes than non-sensitive input features. If we denote input features as $x$, model prediction error as $U$, last-layer representation as $z$, and sensitive attribute classes as $s$, we have $Corr(zU, s) > Corr(x, s)$.*

*Proof.* Proposition 1 simply extends Theorem 1 to the setting of a neural network. We consider a neural network $h$ parametrized by $\theta$ as $h(x; \theta) = \hat{y}$, where $\theta = (W, V)$. Thus, we have $W = (W_1, ..., W_d)^\top \in \mathbb{R}^{D \times M}$ as the weight of

the last layer where $D$ denotes the dimensionality of the last latent representation. $V$ is the weight of all previous layers. $h(x; \theta) = \sigma(W \times z(x; V))$, where $\sigma(z)_j = e^{z_j} / \sum_{d=1}^D e^{z_d}$. The last-layer gradient w.r.t. the cross entropy loss is calculated as

$$\frac{\partial}{\partial W}L(h(x; \theta), y) = z(x; V) \times (\hat{y} - y), \tag{43}$$

where

$$L(h(x; \theta), y) = -\sum_d y_d \cdot \log(h(x; \theta)) \tag{44}$$

$$= \log\left(\sum_{d=1}^D e^{W_d \cdot z(x;V)}\right) - W_y \cdot z(x; V). \tag{45}$$

Note that $\hat{y} - y$ is the bias of the model prediction, which can have a positive/negative value. We define the undirected gradient $g \in \mathbb{R}^{D \times M}$ of the last layer of $h$ by

$$g_{d,j} = z(x)_d|\hat{y}_j - y_j| = z_d U_j, \tag{46}$$

which is the multiplication of the last-layer representation and the prediction error (alternative to model prediction error $U$) of the label class. Here $y_j$ denotes the true value of the $j$-th class in the label.

We can assume that the correlation between the last-layer representation and the label is larger than the correlation between the input features and the label, i.e., $Corr(z, s) > Corr(x, s)$. This assumption is also likely to hold in practice because the purpose of a neural network is to learn a representation to make it easier to predict the label. As long as the neural network is effectively learning representations, this assumption holds. According to Lemma 1, we have

$$\frac{Corr(zU, s)}{Corr(x, s)} > \frac{Corr(xU, s)}{Corr(x, s)}, \tag{47}$$

which means $Corr(zU, s) > Corr(xU, s)$. According to Theorem 1, in general, $I(xU|s) > I(x|s)$. If it works for linear relationships, we have $Corr(xU, s) > Corr(x, s)$. In this case, $Corr(zU, s) > Corr(x, s)$. $\square$

## Acknowledgments

The 2023 CMU CMLH Translational Fellowship in Digital Health supports registration and travel funding.