# OpenReview forum: "Fairness without Demographics on Electronic Health Records"
_AAAI.org/2024/Spring_Symposium_Series/Clinical_FMs — AAAI 2024 SSS on Clinical FMs_

### Official Review · Reviewer_g3NU · 2024-02-16

**Rating:** 8
**Confidence:** 4

**Review:**

This study introduces a weighted loss function based on a gradient graph to improve fairness without needing demographic data. Theoretical results provide the intuition of the method, explaining how the correlation in gradients is related to unfairness. Empirical results show the effectiveness of proposed methods.

Strengths
- Theoretical derivation of method illustrates how it works well
- Empirically, the proposed method yields superior results to other methods

---

### Official Review · Reviewer_Kseb · 2024-02-22
**A new gradient-based technique to boost fairness in healthcare machine learning models without demographic data.**

**Rating:** 8
**Confidence:** 3

**Review:**

Summary
The paper presents a novel method to improve fairness in machine learning models for electronic health records without using demographic data. It constructs a "graph of gradients" to identify underrepresented groups based on model gradient similarity. Weights are learned to increase the exposure of disadvantaged groups. Evaluated on diagnosis prediction tasks, this approach significantly enhances fairness metrics like equalized odds and disparate impact compared to prior algorithms. The paper argues for an interdisciplinary perspective spanning technical, medical, and sociological factors when examining algorithmic fairness in healthcare.

Strength
•	Proposes an innovative graph of gradients method to represent demographic groups and identify underrepresented populations for bias mitigation.
•	Theoretical analysis demonstrates gradients are more closely correlated with demographics than input features under reasonable assumptions.
•	Evaluation of two datasets shows significant improvements in fairness metrics equalized odds and disparate impact over state-of-the-art algorithms.

Weakness
1. The assumptions should be listed before the theorem, not in the appendix
2. The assumption that model accuracy and input features strongly correlate with demographic groups is quite strong and may not always hold in practice. The authors do not provide empirical evidence to support this.
3. The soft grouping method using a graph of gradients is interesting, but the authors do not provide much intuition or analysis into why this works better than prior methods.

---

### Official Review · Reviewer_uUFH · 2024-02-22

**Rating:** 5
**Confidence:** 2

**Review:**

## Summary
This paper attempts to address the important problem of training fair and unbiased Machine Learning (ML) models in clinical settings, when some demographic information about individuals or groups is not available due to privacy or other reasons. To mitigate this, the authors propose to use the model gradients as a method for grouping samples together, based on the intuition that the gradients may carry information about the unknown demographic features. They construct a graph where samples with similar gradients are grouped together, and use it to learn their weights, within the context of an adversarial weighting algorithm. They run experiments on two different medical datasets, and show that their method can improve fairness significantly without much loss in accuracy.

## Strengths
- The idea of using the gradients as a (soft) grouping mechanism, and the intuition that they may be correlated with the unknown demographic features is interesting. The authors also attempt to justify this intuition theoretically, using concepts from information theory.
- The empirical results of their resented adversarial weighting algorithm seem promising, and are competitive against the baselines tested.

## Weaknesses
- In my personal view, the biggest weakness of the paper is clarity: some descriptions are dense, and some parts are hard to follow. The paper would benefit a lot from addressing this. For example, in section 3, notations such as U(h) are suddenly introduced in the text, without explanation. Similarly, in section 4, the method description is not easy to follow. Moreover, it would also benefit the paper to briefly explain the related background material, such as e.g. the Rawlsian Max-Min fairness objective, or some prior methods.
- The experiments presented are on small Neural Networks, although the emphasis of the symposium is on Foundational Models.

 ## Overall Assessment
Overall, I believe the ideas introduced in the paper are valuable, and the empirical results promising. My greatest concern is the clarity of the presentation, and I'm willing to increase my assessment if that is mitigated.

---

### Official Review · Reviewer_deg3 · 2024-02-22

**Rating:** 7
**Confidence:** 4

**Review:**

This paper tackles the challenge of biased predictions in machine learning systems within the clinical domain, where demographic information may be unavailable due to privacy concerns. It introduces an adversarial weighting architecture that utilizes model gradients to identify and prioritize underrepresented groups, offering a novel approach to improving fairness without relying on demographic data. Unlike conventional methods, this approach provides a robust mechanism that significantly enhances fairness while preserving overall accuracy, marking a notable advancement in the quest for fair and reliable machine learning systems in healthcare.